# Microtubule instability driven by longitudinal and lateral strain propagation

Maxim Igaev *, Helmut Grubmüller *

Max Planck Institute for Biophysical Chemistry, Am Fassberg 11, D-37077 Göttingen, Germany

* migaev@mpibpc.mpg.de (MI); hgrubmu@gwdg.de (HG)

**Data Availability Statement:** All relevant data are within the manuscript and its Supporting Information files.

**Funding:** The work was supported by the Max Planck Society (MI and HG) and the German

## Abstract

Tubulin dimers associate longitudinally and laterally to form metastable microtubules (MTs). MT disassembly is preceded by subtle structural changes in tubulin fueled by GTP hydrolysis. These changes render the MT lattice unstable, but it is unclear exactly how they affect lattice energetics and strain. We performed long-time atomistic simulations to interrogate the impacts of GTP hydrolysis on tubulin lattice conformation, lateral inter-dimer interactions, and (non-)local lateral coordination of dimer motions. The simulations suggest that most of the hydrolysis energy is stored in the lattice in the form of longitudinal strain. While not significantly affecting lateral bond stability, the stored elastic energy results in more strongly confined and correlated dynamics of GDP-tubulins, thereby entropically destabilizing the MT lattice.

## Author summary

The dynamic nature of microtubules, long and hollow tubes formed by $\alpha\beta$-tubulin proteins, is crucial for their function is cells, and its precise characterization has been a longstanding problem for cell scientists. Microtubules are essential for cargo transport and provide mechanical forces in chromosome segregation when they disassemble. The disassembly proceeds via changes in the shapes of tubulins upon consumption of a chemical fuel called GTP that binds to every tubulin molecule. This leads to the accumulation of mechanical tension inside the microtubule and ultimately drives it beyond the stability threshold. However, it is still elusive how and where these shape changes contribute to the rapid release of the stored elastic energy. Here, we investigate the behavior of tubulin dimers in a microtubule-like environment using extensive atomistic simulations and show that tubulins locked in the microtubule operate as both 'loadable springs' and 'conformational switches', tightly controlled by their surrounding neighbours. We further show how these shape changes potentially control the overall stability of the microtubule, providing quantitative estimates of the system's energetics.

## Introduction

Microtubules (MTs) are one of the major components of the eukaryotic cytoskeleton and essential for intracellular transport, cell motility, and chromosome separation during mitosis.

Research Foundation via the grant IG 109/1-1 (awarded to MI). The funders had no role in study design, data collection and analysis, decision to publish, or preparation of the manuscript.

**Competing interests:** The authors have declared that no competing interests exist.

These are filamentous assemblies of $\alpha\beta$-tubulin dimers stacked head-to-tail in polar protofilaments (PFs) and folded into hollow tubes via lateral interactions [1, 2] (Fig 1A). Each dimer binds two GTP molecules of which only the one bound to $\beta$-tubulin is hydrolyzed in the MT lattice over time [3, 4]. This hydrolysis reaction is fundamental to MT dynamic instability [5], *i.e.* random switching between phases of growth and shrinkage (Fig 1A). Remarkably, both slow assembly and rapid disassembly of MTs—the latter termed *catastrophe*—are able to perform mechanical work because each tubulin dimer is a storage of chemical energy [6–8].

The switch from a relaxed 'curved' conformation of tubulin favored in solution to a higher-energy 'straight' one is inherent to MT assembly [9–15]. It allows growing MTs to recruit and temporarily stabilize GTP-tubulin in the straight form, most likely due to the greater bending flexibility of GTP-PFs at intra- and inter-dimer interfaces [13, 16–18]. It is therefore conceivable that collapsing MTs would follow a reverse pathway during disassembly; namely, they would release the conformational tension stored in GDP-tubulins that lateral bonds can no longer counteract. However, due to the system complexity and together with the inability of modern structural methods to directly visualize all sequential steps in the GTPase cycle in the straight MT body at high resolutions, it is still unknown exactly how and where the hydrolysis energy is converted to mechanical strain in the lattice.

Recent high- and low-resolution structural studies have revealed, in line with the early finding [19], that the use of a non-hydrolyzable GTP analog, GMPCPP, for MT assembly results in

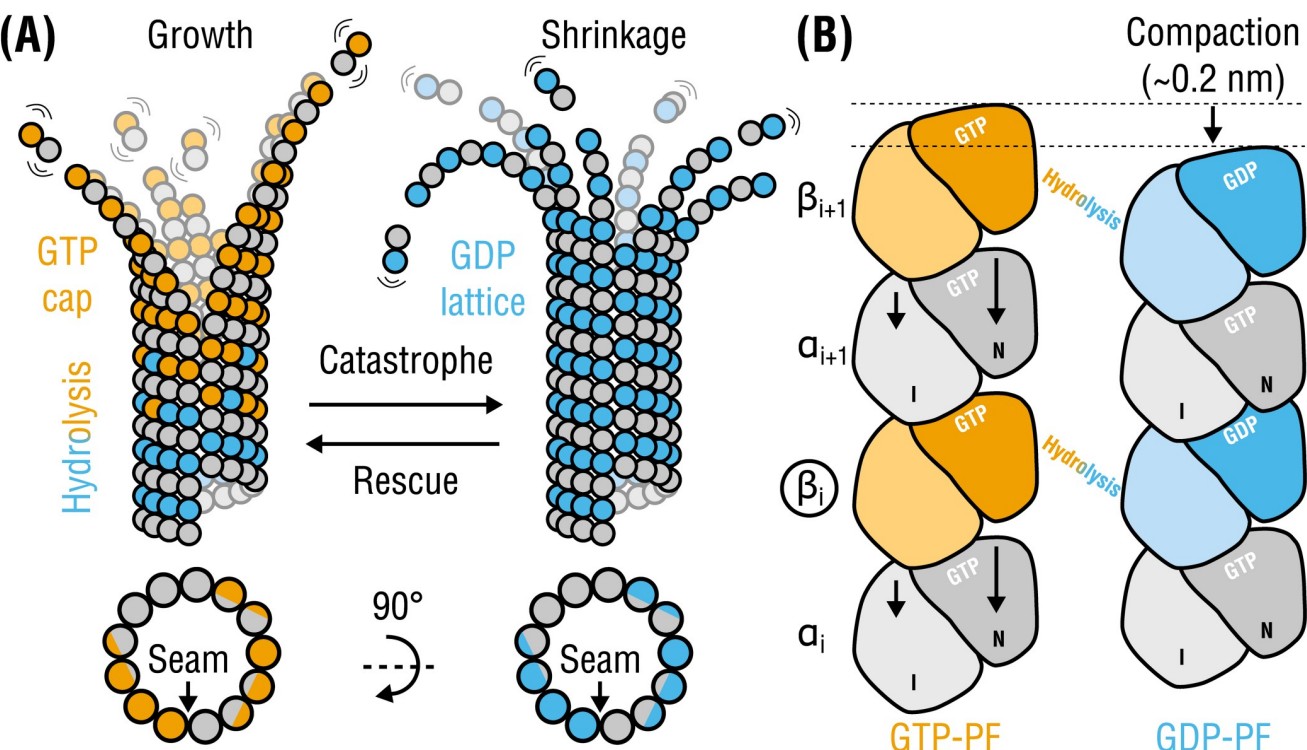

**Fig 1. Tubulin life cycle and lattice compaction upon GTP hydrolysis.** **(A)**, Cartoon representation of structural intermediates in MT assembly and disassembly. Individual dimers are composed of $\alpha$-tubulins (gray circles) and $\beta$-tubulins (orange circles when GTP-bound or cyan circles when GDP-bound). Lattice cross-sections (bottom) indicate the location of the seam interface. **(B)**, Local conformational changes proposed to accompany GTP hydrolysis are shown schematically (viewed from within the lumen). Each monomer is illustrated as two domains: intermediate or I and nucleotide-binding or N (C-terminal domains are not shown for simplicity). Rearrangements in $\alpha$-tubulin around the nucleotide-binding pocket at the inter-dimer interface result in a $\sim 0.2$-nm lattice compaction. The PFs are aligned with respect to monomer $\beta_i$ (marked with a circle). Other more subtle changes (*e.g.*, PF twisting) or intermediate nucleotide states (*e.g.*, GDP-Pi) are not shown for simplicity.

a more expanded MT lattice compared to a fully hydrolyzed GDP-lattice [20–24], which is commonly interpreted as the lattice response to GTP hydrolysis (Fig 1B). Because by itself this global rearrangement does not fully indicate how and whether at all it is linked to GTP hydrolysis and strain accumulation at the single-dimer level, several competing models of MT cap maturation and MT disassembly have been proposed. According to the *seam-centric* or *strain* model [20–22], the gradual build-up of longitudinal tension along the lattice upon GTP hydrolysis is the primary source of MT instability, where the lateral interfaces play only a passive role. In this model, lattice rupture is initiated at the seam because of the greater distance, and presumably weaker interactions, between PFs at this interface observed in unsymmetrized cryo-EM reconstructions. The role of the MT seam as the weakest interface is supported by recent computational evidence [25], but has been challenged experimentally [26]. In contrast, the *holistic* or *bond* model [23, 27] assumes that MT catastrophe can be explained by a sequential weakening of lateral inter-dimer contacts accompanied by a simultaneous strengthening of longitudinal contacts. Also here, recent atomistic simulations of full MT lattices suggest that lateral interactions in GDP-MTs might be weaker than those in GTP-MTs [25]. Finally, the most recent '*no expansion*' model [24] provides an alternative view of the cap maturation process in which both pure GTP- and pure GDP-MTs have equally compacted lattices, while the higher-energy expanded lattice induced by GTP hydrolysis (mimicked by GMPCPP) corresponds to an intermediate, phosphate-releasing state. This model is partially supported by the observation that the extent of lattice compaction differs across different eukaryotic species [28].

The coexistence of the three models originates from the fact that the interplay between tubulin intrinsic strain and lateral binding inside the straight MT body is largely unclear. Indeed, the subtle changes in lattice compaction and dimer-dimer contacts would be best studied within straight PF assemblies in the presence or absence of lateral neighbors and conditioned on a fixed nucleotide state, which has not yet been achieved. This has prompted us to assess the mechanochemistry of both lattice compaction and lateral inter-dimer coupling using extensive molecular dynamics (MD) simulations of (i) isolated PFs, (ii) standard (homotypic) double-PF systems, as well as (iii) three-PF lattice patches. In all cases the PFs are locked in the straight conformation due to the use of periodic boundaries along the MT axis. Essentially and by construction, this setup allows to probe both mechanics and lateral cooperativity of *individual* dimers embedded in straight lattice regions distant from the dynamic tip. Because the three models of MT disassembly assume different properties of assembled tubulins and their lateral interactions, it is hence possible to test all models directly. By focusing on small, controllable MT-like subsystems, our simulations provide new insights into the lattice mechanics and energetics that drive MT disassembly.

## Results

### GTP hydrolysis in *β*-tubulin stiffens individual PFs

If MTs accumulate longitudinal elastic strain upon GTP hydrolysis, one would expect them to change the mechanical properties of individuals PFs also in the absence of lateral interactions. We therefore asked how the nucleotide state affects both equilibrium conformation and elasticity of isolated PFs and how much mechanical energy can be potentially stored in a single dimer upon GTP hydrolysis. The recent cryo-EM reconstructions of MTs in non-hydrolyzable GMPCPP- (mimicking GTP) and GDP-state [21, 22] enabled us to construct atomistic models of isolated PFs (Fig 2A) using correlation-driven molecular dynamics [31] and to assess their equilibrium and elastic properties by atomistic MD simulations (see Methods for details on system preparation, cryo-EM model refinement, simulation protocol).

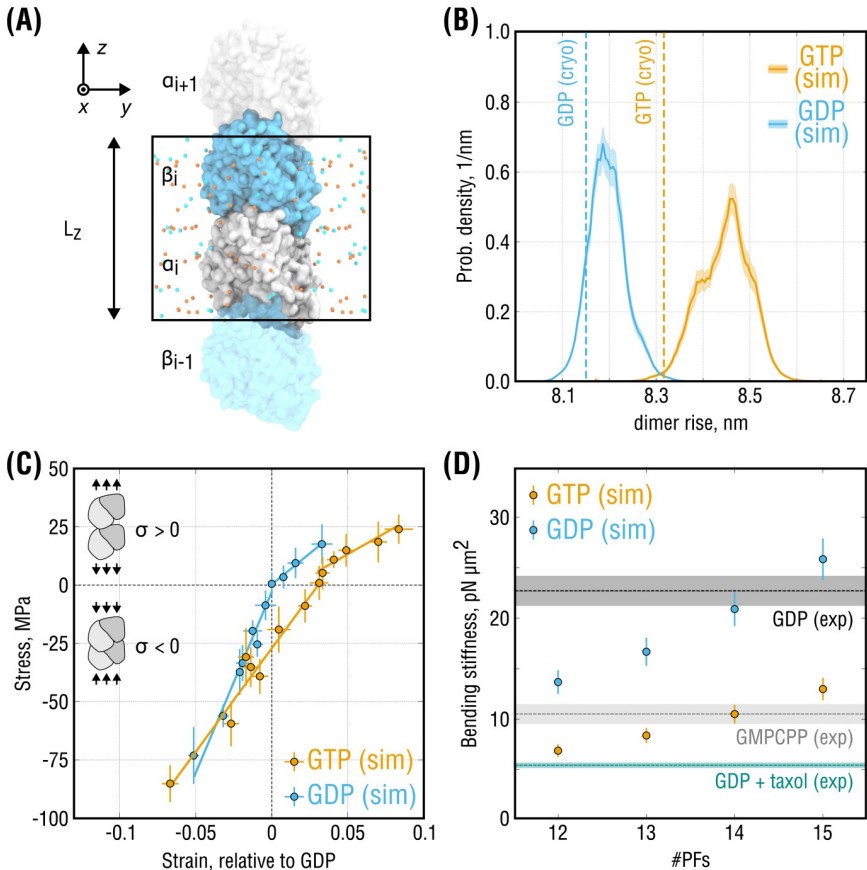

**Fig 2. Elastic properties of isolated 'infinite' PFs. (A)**, Simulation setup for the single-PF system. $\alpha$-tubulin (gray) and $\beta$-tubulin (cyan) are shown in surface representation. Potassium and chloride ions are shown as orange and cyan spheres, respectively. Water molecules are hidden for clarity. Periodic box with the axial dimension $L_z$ is marked by a black rectangle. **(B)**, Equilibrium probability distributions of the dimer rise in the PFs obtained from stress-free simulations of the system in **(A)**. Shaded areas show statistical uncertainties of the distributions estimated with umbrella sampling. Dashed lines indicate the dimer rise values observed in the cryo-EM densities of GMPCPP- and GDP-MTs. **(C)**, Stress-strain curves calculated for the system in **(A)** in both GTP- (orange) and GDP-state (cyan). Strain is computed relative to the equilibrium dimer length of GDP-PF, and negative (positive) stresses correspond to PF compression (extension). Only separate fits to the positive and negative stress ranges are shown. **(D)**, Bending stiffness parameters of GTP- and GDP-MTs calculated using the elastic moduli in **(B)** (all stress values) and for varying PF numbers (orange and cyan dots, respectively). Experimental values (dashed lines with shaded areas) represent inverse-variance weighted means and standard deviations that combine multiple independent thermal fluctuation measurements summarized in [29] and recently updated in [30].

To assess the equilibrium properties of isolated PFs at room temperature, we first performed multiple simulations of GTP- and GDP-PFs totaling $\sim 23\ \mu$s and $\sim 13\ \mu$s, respectively, following a previously published protocol [32]. We monitored the dynamics of both tubulin dimer shape and axial periodic box size $L_z$, *i.e.* the lattice spacing (Fig 2A). There are three possibilities how the conformation of the PF could contribute to an increase or decrease of $L_z$: (a) changes at the intra-dimer interface between $\alpha$- and $\beta$-subunits belonging to the same dimer (referred to as 'dimer spacing'), (b) changes at the inter-dimer interface between $\alpha$- and $\beta$-subunits belonging to neighboring dimers along the PF axis (referred to as 'PF spacing'), and (c) changes in the shapes of $\alpha$- and $\beta$-subunits due to elastic deformations. We then employed the Functional Mode Analysis [33, 34] to train a regression model on the dynamics of $L_z$ and to derive a reaction coordinate that best describes the compaction/expansion dynamics of the PF

in equilibrium (see Methods). This reaction coordinate, that we termed '*dimer rise*' for consistency with cryo-EM experiments, was used in all subsequent free energy calculations.

Fig 2B shows the equilibrium probability distributions of the dimer rise as a function of the nucleotide state computed with additional umbrella sampling simulations ($\sim$114 $\mu$s of cumulative simulation time; see Methods). Both GTP- and GDP-PFs slightly increased the lattice period during the simulations relative to their initial cryo-EM conformations. This slight elongation might be caused by thermal expansion of the 70–80 K cryo-EM structures after re-equilibration at room temperature. This possibility is supported by the observation that the relative elongation is largely independent of system size and sampling, as will be seen further. However, GTP-PFs maintained a significantly longer dimer rise compared with GDP-PFs (+0.25 ± 0.07 nm), consistent with the experimentally observed difference of $\sim$0.2 nm. In the following, we will refer to these two states as *expanded* and *compacted*. Hence, it is likely that the difference between the global states of MT structure seen by cryo-EM reflects a local response of tubulin to GTP hydrolysis; otherwise, it would vanish in the absence of lateral contacts.

Further, GTP-PFs sampled a wider range of dimer rise values as indicated by the distribution widths in Fig 2B, which suggests that GTP-PFs are mechanically more flexible. We previously showed that, when placed in solution, GTP-tubulin exhibits higher bending flexibility than GDP-tubulin [17]. It was therefore surprising that, when tubulin was locked in the straight MT-like conformation, also the longitudinal elasticity of the dimer was affected by the nucleotide state.

To quantify the mechanical elasticity of the system in Fig 2A, we performed a set of steady-state compression/extension simulations at constant values of the axial component $P_{zz}$ of the pressure tensor (along the PF). Fig 2C shows the obtained strain-stress curves, where the strain was computed relative to the equilibrium conformation of GDP-PFs. Irrespective of the nucleotide state, the stress-strain data clearly falls into two elastic regimes: a rather soft response for positive stresses (extension) and a much stiffer response for negative stresses (compression). This previously observed behavior of GDP-tubulin [32], which we here confirmed using higher quality structures of both nucleotide states and wider strain ranges, emerges likely because different parts of the heterodimer are involved in the mechanical response upon compression or extension. Whereas extension mainly stretches the inter-dimer and, to a lesser extent, intra-dimer interfaces, compression forces individual monomers to change their shapes, causing much more resistance. We therefore analyzed the positive and negative strain ranges separately.

A linear fit to the negative stress data of the GTP- and GDP-PF simulations yielded elastic moduli of 0.89 ± 0.07 GPa and 1.77 ± 0.13 GPa, respectively. Fitting to the positive stress data yielded systematically smaller moduli of 0.37 ± 0.07 GPa (GTP-PF) and 0.53 ± 0.03 GPa (GDP-PF). A fit to the entire stress range analyzed in our simulations resulted in values of 0.82 ± 0.06 GPa (GTP-PF) and 1.55 ± 0.15 GPa (GDP-PF) that agreed better with the moduli obtained by fitting the negative stress data only. Whereas the single-PF system tolerated high compression stresses up to $P_{zz}$ = +200 bar without undergoing plastic deformations and irrespective of the nucleotide state, this was not the case for extension stresses. GTP-PFs withstood stretching up to $P_{zz}$ = −65 bar without rupturing at the inter-dimer interface in the course of our simulations ($\sim$1 $\mu$s each). In contrast, GDP-PFs ruptured already at stress values below $P_{zz}$ = −40 bar, implying that a lower force is likely sufficient to break the longitudinal bond. Although more sampling would be required to investigate PF rupture pathways, our stress-strain data (Fig 2C) together with the equilibrium free energy calculations (Fig 2B) support the interpretation that straight GDP-PFs are stiffer than GTP-PFs while they might possess more fragile inter-dimer longitudinal bonds.

The factor of two difference in the elastic moduli of GTP- and GDP-PFs is remarkable given the high structural similarity of the two conformational states. Careful review of existing measurements of MT bending mechanics reveals that, despite variations in the experimental protocols and theoretical models used to analyze such data, MTs are intrinsically softer when polymerized in the presence of GMPCPP and/or Taxol [29, 30]. This is reflected, *e.g.*, in a significantly distinct bending stiffness $E \times I$, where $E$ is the elastic modulus and $I$ is the second moment of the cross-sectional area of the MT. We therefore asked if the nucleotide-dependent elasticity of PFs (Fig 2B and 2C) might explain the experimentally observed differences in coarse-grained elastic properties of MTs.

Bending stiffness of MTs is typically obtained by monitoring and quantifying their equilibrium fluctuations or by directly applying a force to bend MTs and then measure their resistance (e.g., by using optical tweezers) [29]. It is known that in thermal fluctuation experiments, MTs behave on average stiffer than in force-probing experiments [29, 30]. It has been proposed that this discrepancy can be reconciled by taking into account that large deformations caused by external forces acting on MTs could surpass the elastic limit, which would lead to non-elastic deformations of tubulin dimers and/or breakage of inter-dimer contacts [30]. Because such events are, by construction, unlikely to happen in our simulation setup, we compared our results only with equilibrium fluctuation experiments, where tubulin dimers are mostly subject to small-strain elastic deformations.

Fig 2D compares bending stiffnesses of hypothetical MTs with varying PF numbers calculated using our data in Fig 2C (all stress values) and consensus values calculated as precision-weighted averages from a pool of independent experimental measurements (reviewed in [29] and recently updated in [30]; see Methods). The comparison revealed that a good agreement between the experimental data and our calculations can only be achieved if the PF number is approximately 14 for both GTP- and GDP-MTs. It is known that MT mechanics is highly sensitive to changes in the PF number (see Discussion in [35]). Most MTs polymerized in vitro without co-factors and MT-binding drugs possess 14 PFs [36], with ratios of 13-PF to 14-PF MTs reaching approximately 1:9 for GDP-MTs and 1:3 for GMPCPP-MTs [21, 22]. Assuming that tubulin axial elasticity does not depend on the PF number, the two-fold higher bending stiffness of GDP-MTs can be accounted for almost entirely by a two-fold higher elastic modulus of GDP-PFs, at least for small-strain deformations.

Finally, the good agreement of our elasticity calculations with experimental knowledge allowed us to estimate that a free energy of $\Delta G_{el} \approx 11.6\,k_B T$, where $k_B$ is the Boltzmann constant, would be stored in a GTP-PF per dimer when mechanically compressed to the state of a GDP-PF (see S1 Material). Remarkably, this energy is very close to both the energy harvested by MTs upon GTP hydrolysis [19, 37] and the maximal excess energy that can be stored in an MT lattice to maintain one of the most favorable configurations ($\sim 11\,k_B T$ per dimer for MTs with 13 or 14 PFs [38–40]). Together with the consistency of our calculated elastic moduli with the observed softening of GMPCPP-MTs or Taxol-stabilized MTs *vs*. GDP-MTs, this strongly suggests that almost the entire energy available from GTP hydrolysis is stored in the MT lattice in the form of longitudinal elastic strain. We note that, during model preparation, the GMPCPP molecules were manually converted to GTP, and the starting structures were allowed to adapt to this change in the subsequent production simulations (see Methods). However, it cannot be ruled out that some effects of GMPCPP on the dimer conformation may still remain. Nevertheless, in the absence of alternative high-resolution models of the putative GTP state, we consider our tubulin model a sufficiently good approximation of the true GTP-tubulin structure in MTs.

### Lateral coupling and GTP hydrolysis reduce conformational freedom of tubulin in PFs

One of the key unanswered questions is how the MT lattice would accommodate laterally coupled dimers in conflicting conformational states (expanded *vs.* compacted), a situation that is very likely to arise downstream from the growing MT tip. It was previously speculated that such a structural conflict would either weaken the lateral interactions between incompatible dimers or increase the rate of GTPase activity [41, 42]. In the latter case, the hydrolysis-triggered compaction of an expanded dimer located next to a compacted dimer would be more favorable. However, testing these hypotheses experimentally is currently challenging.

To get insight into how the presence and conformation of a lateral neighbor affects the compaction-expansion dynamics of tubulin in PFs, we constructed atomistic models of double-PF systems in both nucleotide states (Fig 3A; see Methods for model refinement and simulation protocol). We then computed free energy surfaces of the double-PF systems as a function of dimer rise and nucleotide state using the umbrella sampling approach with $\sim 80$ $\mu$s of cumulative simulation time (Fig 3B and 3C; see Figure A in S1 Material for statistical uncertainties). Like the isolated PFs (Fig 2B), the double-PF systems adopted, on average, slightly more expanded conformations relative to their starting cryo-EM structures, most likely, due to thermal expansion. Also, the constant shift between the two distributions by $0.19 \pm 0.05$ nm was preserved, which was close to the experimentally observed difference of $\sim 0.2$ nm and, within statistical error, consistent with the difference of $0.25 \pm 0.07$ nm calculated for the isolated PFs (Fig 2B).

As described above, one would expect each PF in the double-PF system to behave differently depending on both conformational state of the neighbor and own nucleotide state. In particular, their motion should be statistically correlated due to lateral coupling. In addition, the substantial difference in mechanical flexibility of isolated GTP- and GDP-PFs (Fig 2B and 2C) should be reflected in the dynamics of coupled PFs as well. To test these expectations, we

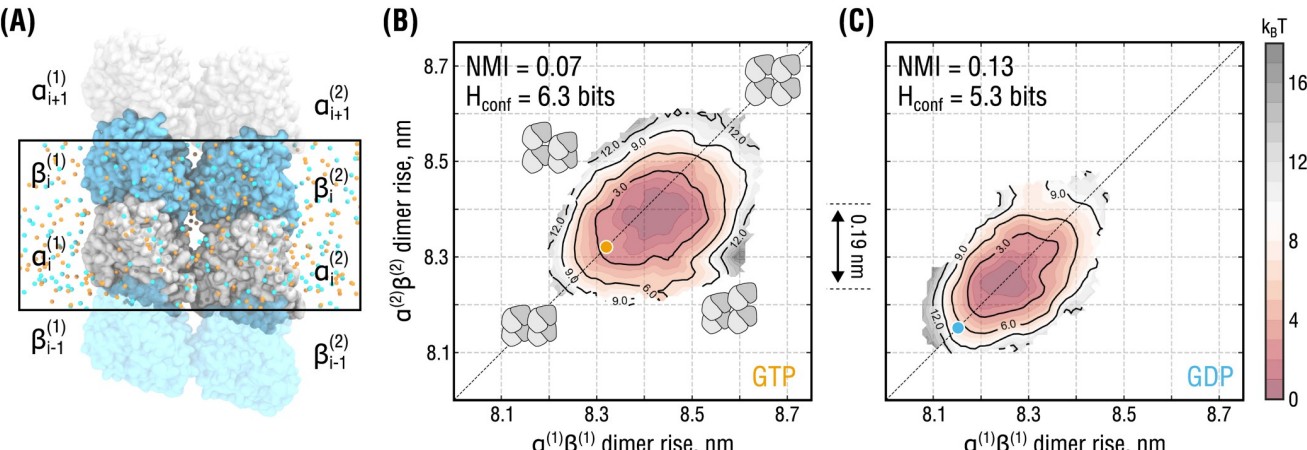

**Fig 3. Lateral coupling and nucleotide state affect PF dynamics. (A)**, Simulation setup for the double-PF system mimicking a standard (homotypic) lateral interface. Color coding as in Fig 2A. Water molecules are hidden for clarity. Periodic box is marked by a black rectangle. Individual PFs are labeled as (1) and (2). **(B)** and **(C)**, Free energy surfaces of the system in **(A)** as a function of dimer rise and nucleotide state obtained by umbrella sampling. The surfaces are color-coded by free energy values with an increment of 1 $k_BT$ (dark red to gray). Black solid lines additionally show isoenergetic contours. Orange and cyan circles indicate the dimer rise values observed in the cryo-EM densities of GMPCPP- and GDP-MTs, respectively. Cartooned dimers in **(B)** schematically show the extreme conformations of the double-PF system in which both are similarly expanded or compacted (along the diagonal) or in conflicting conformations (along the anti-diagonal). The relative shift of 0.19 nm between the minima of the free energy surfaces in **(B)** and **(C)** is additionally indicated.

introduced two metrics to quantify the changes in the double-PF free energy profiles upon nucleotide exchange. First, we used the *normalized mutual information* (NMI) which is a mere statistical measure of both linear and nonlinear correlation between two stochastic variables (see Methods for the rigorous definition). For the particular system in Fig 3A, NMI would be zero if the PFs moved fully independently or unity if their dimer rise fluctuations were fully synchronized. Second, we used the *confinement entropy* $H_{conf}$ that quantifies the conformational space 'volume' available to both PFs, irrespective of how much the PF motions are correlated (see Methods for the rigorous definition). For the particular system in Fig 3A, $H_{conf}$ would be close to zero if the dimer rise fluctuations were strongly localized around fixed values or maximal if all dimer rise values were equally likely. Both stronger inter-PF correlation (higher NMI) and higher PF stiffness (lower $H_{conf}$) would naturally result into the PFs having a more restrictive influence on each other, moving the double-PF system up in free energy due to the associated loss of conformational entropy. Vice versa, the joint conformational space increases when the PFs become more flexible and their fluctuations become less correlated, hence moving the double-PF system down in free energy.

As visible from the free energy surfaces having elliptic shapes extended along the diagonal (Fig 3B and 3C) and supported by the calculated NMI values, the conformations of the double-PF system in which the PFs were similarly expanded or compacted were lower in free energy than those in which the PFs adopted conflicting conformations. Thus, the PFs have a mutually restrictive influence on each other, penalizing configurations in which the PF conformations are too different. As a result, the double-PF system exhibits more correlated dynamics than would be the case if the PFs were isolated. Furthermore, the correlation and confinement effect was stronger for the system in GDP-state ($\Delta NMI = +0.06$ and $\Delta H_{conf} = -1.0$ bits compared to GTP-state). Together with the PF stiffening upon GTP hydrolysis (Fig 2B and 2C), this suggests that GTP hydrolysis further reduces the conformational space available to tubulin dimers in the double-PF system, making it thermodynamically less favorable than that in GTP-state.

## Nearest-neighbor interactions between PFs modulate GTPase response of tubulin

Our observation that the double-PF system favors conformations in which the PFs are similarly expanded/compacted suggests that the system is less stable when there is a conformational mismatch between the PFs, likely because the lateral bond would be under excessive shear tension. To quantify the extent of lateral bond destabilization by the conformational mismatch between the PFs, we considered a thermodynamic cycle shown in Fig 4A, following a previous scheme [43]. We assume that the equilibrium conformation of the double-PF system can be changed into the one with a conformational mismatch between the PFs (vertical transitions in Fig 4A). The free energy cost associated with this transformations ($\Delta G_{eq\to mis}$) was calculated using our previous umbrella sampling results for both single- and double-PF system (Figs 2B and 3B and 3C; see S1 Material). Because the sum over all transition paths in the cycle must vanish, the difference between these values, $\Delta G_{eq\to mis}^{double} - \Delta G_{eq\to mis}^{single}$, equals the bond stability of the mismatched double-PF system relative to the equilibrium case, $\Delta\Delta G^{assoc} = \Delta G_{mis}^{assoc} - \Delta G_{eq}^{assoc}$ (horizontal transitions in Fig 4A). Hence, a positive $\Delta\Delta G^{assoc}$ equally implies that (a) the PF association is less favorable when the PFs are in conflicting compaction states, or that (b) the GTPase response of an expanded dimer is stimulated if its nearest neighbor is in the compacted state.

Fig 4B and 4C shows $\Delta\Delta G^{assoc}$ relative to the lowest-energy system configuration (free energy minima in Fig 3B and 3C, respectively) as a function of dimer rise and nucleotide state.

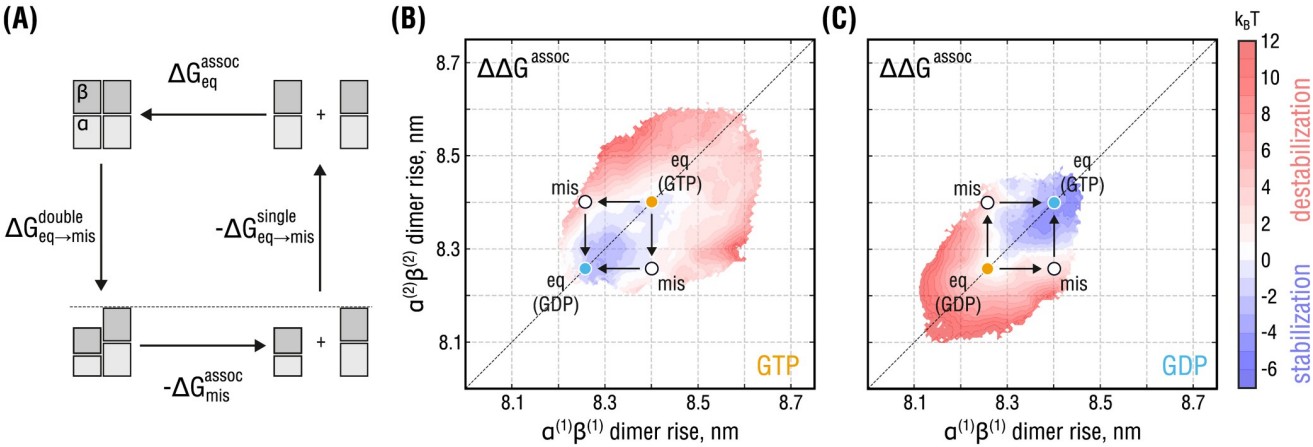

**Fig 4. Relative thermodynamic stability of the lateral bond in the double-PF system.** (**A**), Thermodynamic cycle demonstrating the idea behind estimating the effect of unequal PF conformations on the association free energy between the PFs. While simulating the horizontal transitions (PF association) is computationally more expensive, the free energy changes linked to the vertical transitions (PF compaction) have already been obtained (Figs 2 and 3). (**B**) and (**C**), Distributions of the relative stability of the double-PF systems with respect to their equilibrium conformations marked with orange and cyan circles for GTP- and GDP-state, respectively, as a function of dimer rise and nucleotide state. White circles denote conformations with the strongest observed dimer rise mismatch. Free energy color coding is adjusted such that red (blue) areas correspond to conformations of the double-PF system in which the lateral bond is destabilized (stabilized) relative to equilibrium. White areas correspond to no change in the lateral bond stability.

The calculations suggest that a conformational mismatch between the PFs would have a statistically significant effect on the thermodynamic stability of the double-PF system, corresponding to a change of $\Delta\Delta G^{assoc} = +4.0 \pm 1.6 \, k_BT$ (equilibrium constant fold-change by $\sim 55$). In contrast, simultaneous compaction/expansion of the two PFs has no statistically significant effect on the stability of the double-PF system with a relative change of $\Delta\Delta G^{assoc} = -1.0 \pm 1.5 \, k_BT$ (equilibrium constant fold-change by $\sim 0.37$), implying that the lateral bond is stabilized once the conformational mismatch is resolved. Our results, therefore, provide quantitative evidence for the previous ideas that a structural conflict at the lateral interface due to unequal nucleotide states would either weaken it or locally increase the rate of GTPase activity [41, 42], *i.e.* locally facilitate the compaction transition. However, not only do we propose that both ideas would be equivalent, but we also estimate the magnitude of lateral bond destabilization and predict that it would be a transient and reversible effect. Our estimate for the bond destabilization energy in the absence of any lateral mismatch, $\Delta\Delta G^{assoc} = -1.0 \pm 1.5 \, k_BT$, also agrees well with a recent computational study by the Odde lab [44], where only a weak nucleotide dependence of the lateral bond stability was found, though using a finite PF setup.

We note that the association free energy differences in Fig 4A (representing lateral transitions) refer to a situation in which one long and straight PF fully associates with another. The considered scheme differs from how PFs most likely associate/dissociate at the dynamic MT tip, namely, dimer by dimer while bending away from the MT lumen. Hence, $\Delta G^{assoc}_{mis}$ and $\Delta G^{assoc}_{eq}$ describe per-dimer contributions to the lateral thermodynamic stability of MT lattices in regions distant from the dynamic MT tip.

## Nearest-neighbor interactions between PFs cause long-range correlations in the lattice

The finding that lateral coupling leads to more confined and correlated dynamics of tubulin in the double-PF system is explained by the nearest-neighbor interaction that prevents dimers in the adjacent PFs from adopting conflicting conformations by energetically penalizing local

mismatches. It is therefore clear that also the motions of dimers situated in distant PFs should be correlated as a consequence of the elementary short-range interactions shown in Fig 3B and 3C. However, it is unclear to what extent the nucleotide state would affect such *long-range correlations*. To quantify their magnitude and the dependence on the bound nucleotide in similar minimalist but computationally feasible settings, we constructed a larger PF system comprising $3 \times 1$ dimers per periodic box dimension $L_z$ (Fig 5A and 5B), which allowed us to quantify the statistical correlation between a pair of non-adjacent PFs. From the equilibrium dynamics of this three-PF system, similarly as above, we estimated its free energy landscape as a function of dimer rise and nucleotide state and subsequently disentangled nearest-neighbor interactions and long-range correlations.

As it was unfeasible to perform sufficiently accurate free energy calculations for such a large system, we instead resorted to a Bayesian inference approach that integrates prior knowledge about the energetics of the smaller subsystems (Figs 2A and 3A) to infer the joint free energy distribution of the three-PF system from unbiased MD simulations (see S1 Material). To this end, six independent, 600-ns long equilibrium simulations of the three-PF system in each nucleotide state were performed, yielding a total of $\sim 7.2 \ \mu s$ of sampling time. The inferred three-dimensional (3D) joint free energy distributions were then pairwise projected onto planes corresponding to two-dimensional (2D) free energy landscapes of adjacent and non-adjacent double-PF subsystems (Fig 5C and 5D). Consistent with the single-PF and double-PF systems analyzed above (Figs 2 and 3), the conformation of the three-PF system in our simulations was more expanded than the underlying cryo-EM structures, while the nucleotide-dependent difference in lattice compaction, again, was preserved.

The NMI values for the non-adjacent free energy landscapes were calculated ($NMI_{13}$) and compared with those for the adjacent landscapes in the same system ($NMI_{12}$ and $NMI_{23}$). If the non-adjacent PFs did not interfere, $NMI_{13}$ would be negligible relative to both $NMI_{12}$ and $NMI_{23}$, yielding almost circular free energy landscapes in Fig 5C and 5D (center). However, we found that $NMI_{13}$ is only by a factor $\sim 0.5$ and $\sim 0.85$ smaller than the values for the directly interacting PFs in GTP- and GDP-state, respectively. This suggests that the correlations between non-adjacent PFs induced by the nearest-neighbor PF interactions are enhanced upon GTP hydrolysis.

In fact, several recent findings provide intriguing evidence that weaker intra-lattice correlations might stabilize the MT. First, some MT-stabilizing drugs such as Taxol have been recently shown to increase the lattice heterogeneity of GDP-MTs as compared to drug-free GDP-MTs [45], which resonates with the ability of Taxol to restore the bending flexibility of GDP-MTs [46, 47]. Second, a very similar effect on MT stability and mechanical resilience has been reported for acetylated *vs.* wild-type MTs [48, 49], likely due to a small but additive allosteric effect of $\alpha$-tubulin acetylation at residue K40 [50]. In light of our drug- and acetylation-free simulation results, we propose that GTP hydrolysis reduces tubulin axial flexibility and enhances short- and long-range correlations between PFs, thereby leading to a loss of conformational entropy by the MT lattice.

## Discussion

Tubulin dimers locked in the MT lattice operate as 'loadable springs' and 'conformational switches' whose load and conformation critically depend on both nucleotide state and lattice surrounding. The result is a metastable behavior of MTs because they have to reconcile the favorable dimer-to-lattice binding and the internal strain build-up that is fueled by GTP hydrolysis and has a destabilizing effect. Not surprisingly, it is hard to reach a consensus on

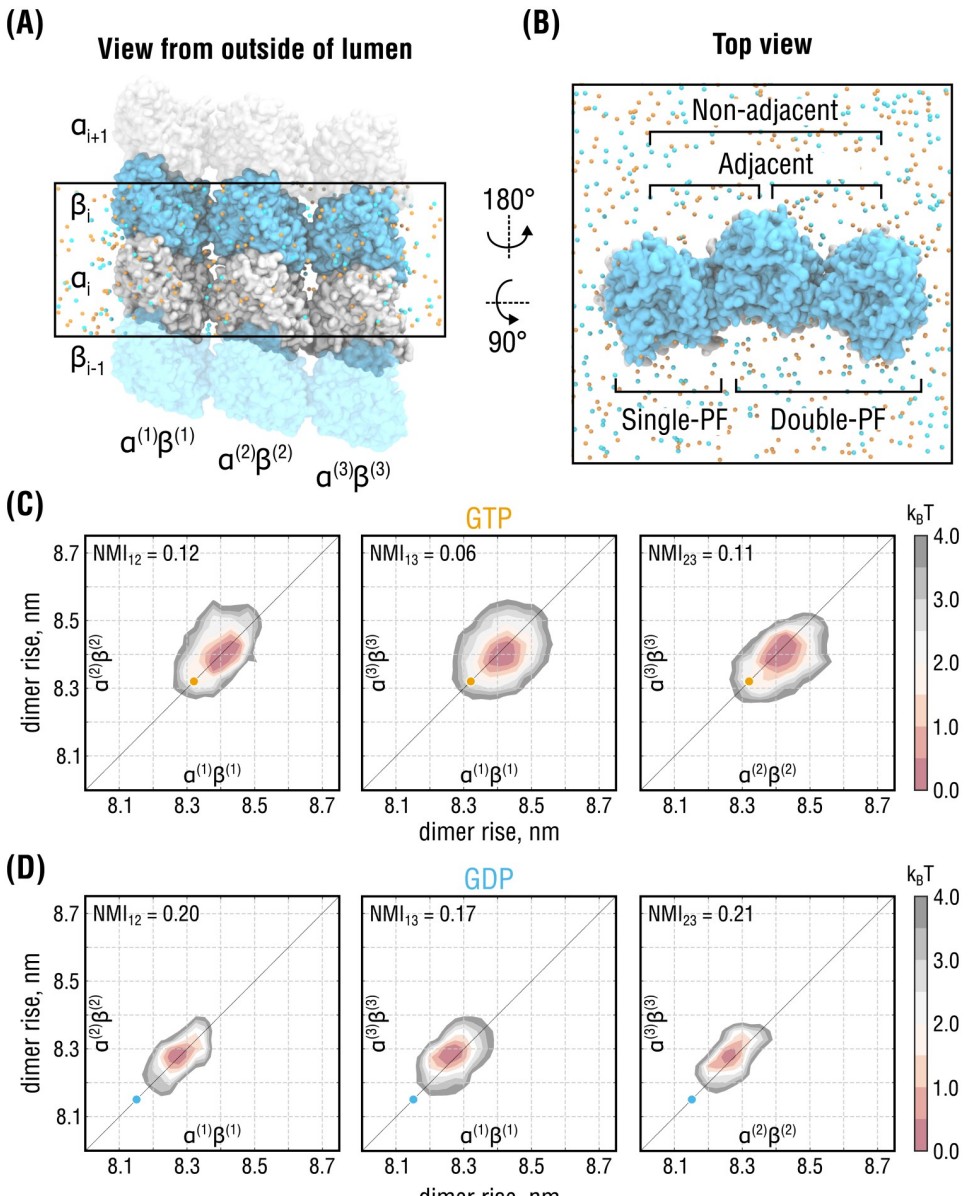

**Fig 5. Lateral coupling induces long-range correlations between distant PFs. (A)** and **(B)**, Side and top views of the simulation setup for the three-PF system mimicking a larger segment of the MT lattice. Color coding as in Figs 2A and 3A. Water molecules are hidden for clarity. Individual PFs are labeled as (1), (2) and (3). **(C)** and **(D)**, Free energy landscapes of the system in **(A)** as a function of dimer rise and nucleotide state. The 3D landscapes were pairwise projected onto planes corresponding to 2D free energy landscapes of adjacent ($\alpha^{(1)}\beta^{(1)} - \alpha^{(2)}\beta^{(2)}$ and $\alpha^{(2)}\beta^{(2)} - \alpha^{(3)}\beta^{(3)}$, left and right, respectively) and non-adjacent PFs ($\alpha^{(1)}\beta^{(1)} - \alpha^{(3)}\beta^{(3)}$, center). Orange and cyan circles indicate the dimer rise values observed in the cryo-EM densities of GMPCPP- and GDP-MTs, respectively. Note the shift between the GTP and GDP distributions by $\sim 0.2$ nm along both reaction coordinates, consistent with the other simulations in Figs 2 and 3.

precisely where this chemical energy is converted to mechanical strain and spread over the lattice because the system complexity allows various interpretations.

To clarify this issue, we have aimed at a quantitative understanding of the interplay between tubulin intrinsic strain and lateral binding inside straight MT-like compartments. Our results support the following conclusions: (i) there is a two-fold increase in longitudinal lattice tension

upon nucleotide hydrolysis, which we attribute to the increased stiffness of GDP-PFs; (ii) lateral coupling between PFs reduces the conformational flexibility of tubulin by entropically penalizing PF conformations that are too different; (iii) restrictive interactions between neighboring PFs induce long-range correlated motions of non-adjacent PFs; (iv) both short- and long-range cooperativity of PF motions is stronger for GDP-PFs suggesting a loss of conformational entropy by MT lattices upon GTP hydrolysis.

Our computational findings provide quantitative insights into tubulin mechanochemistry and, therefore, the structural and energetic basis of MT dynamic instability. The results presented here enable us to test the three major models of MT cap maturation and MT catastrophe: (i) the strain model proposed by Nogales, Alushin, Zhang and colleagues [20–22], (ii) the bond model proposed by the Moores lab [23, 27], and (iii) the most recent 'no expansion' model by Estévez-Gallego et al. [24]. Our results do not support the bond model because we do not find a statistically significant effect of the nucleotide state on the lateral bond stability, which is key to the bond model. This has also been confirmed by an earlier computational study from the Odde lab using a finite PF setup [44]. Rather, our results are consistent with a weakening of longitudinal bonds upon GTP hydrolysis, based on the stronger response of single GDP-PFs to mechanical stretching (Fig 2C), which is indicative of more fragile longitudinal bonds between GDP dimers. Further, our results do not support the most recent model by Estévez-Gallego et al. [24] postulating that both pre- and post-hydrolysis MT segments are equally compact, while the lattice undergoes a transient, hydrolysis-energy-consuming expansion (approximated by the GMPCPP lattice) to release the $\gamma$-phosphate. Our combined evidence (Figs 2–5) rather suggests: (a) whatever the GMPCPP state corresponds to in the GTPase cycle of tubulin, it most likely precedes the GDP state; and (b) a pure GDP lattice is higher in free energy than a pure GMPCPP one because the transition to the GDP lattice can only be achieved by investing a per-dimer energy on the order of 11 $k_BT$. Thus, the conformational cycle proposed by Estévez-Gallego et al. would involve around twice the energy available from GTP hydrolysis to first expand and then compact the lattice, which is energetically implausible according to our estimates.

Overall, our results are currently most consistent with the strain model by Nogales, Alushin, Zhang et al [20–22]. However, until the status of the GMPCPP lattice is entirely clear, the strain model remains incomplete. This model does not describe the behavior of mixed nucleotide MT lattices, which we have now predicted using long-time MD simulations and free energy calculations. At present, the strain model cannot preclude the possibility of tubulin adopting unknown pre-hydrolysis conformations prior to that mimicked by GMPCPP. Although our simulations do not show large rearrangements of the GMPCPP-tubulin structure upon replacement of GMPCPP with GTP, we still have to assume that the GMPCPP-MT lattice is sufficiently similar to the unknown pre-hydrolysis GTP-MT lattice. Whatever the precise conformational cycle, our results agree best with the view of GMPCPP-tubulin being *one* of the cap-stabilizing, expanded, and flexible conformations that are unlikely to be preceded by stiffer and compacted ones. Perhaps a more promising approach toward ultimately resolving this issue in future structure determination efforts would be to use knowledge-based point mutations that selectively uncouple the tubulin conformational and GTPase cycles [51, 52].

Taken together, a new picture emerges in which the MT lattice stability is not exclusively determined by the nucleotide-dependent dynamics of individual dimers, but more generally, by their non-additive collective behavior. In this work, we provide a thermodynamic explanation for the intrinsic destabilization (distant from the dynamic tip) which precedes MT breakdown and which relies on the idea that MT lattices gradually accumulate mechanical strain and lose conformational entropy as GTP hydrolysis proceeds. In other words, the MT becomes thermodynamically less and less stable already during the growing phase, which predisposes it

to explosive strain release. Exascale atomistic simulations ($\gg 10^6$ atoms) and coarse-grained kinetic models can now be used to extrapolate how the results of our study will apply to the time evolution of the MT plus-end tip at much larger spatiotemporal scales.

By construction, our lattice simulations focus on straight 'infinite' PF systems with only one dimer layer per periodic box length. As a result, each dimer interacts with itself along the MT axis. Such setups are established and well-tested in the MD field, and the resulting artifacts are well-characterised [32, 53–55]. For the periodic tubulin systems at hand, two types of artifacts warrant attention. First, fluctuations with wavelengths larger than the box size are suppressed and, therefore, fluctuations of the dimer rise may be smaller than in a simulation with a much larger box size or in reality. Second, 'diagonal' correlations are not present in the double- and three-PF systems, *i.e.* a dimer is unable to influence the conformations of other dimers in the neighboring PFs located in the layer above or below that dimer. Another possible issue is the fact that our simulations did not include a closed segment of the MT body with 14 PFs so that possible 'edge' effects cannot be fully discounted. Contrary to the periodic boundaries, this simplification of the MT geometry might lead to more relaxed fluctuations of the dimer rise for those PFs having only a single neighbor. It is conceivable that, to some extent, the periodic boundaries mitigate the absence a closed MT lattice due to error cancellation.

It is therefore justified to ask what the simulated PF models actually represent in a real MT system and how strongly the chosen simulation setup affects the conclusions of our study. Two observations are important here. First, our PF elasticity calculations provide estimates that are largely consistent with previous experimental knowledge, indicating that longitudinal PF mechanics does not significantly depend on the choice of simulation protocol. Second, we primarily focus on the effect of the nucleotide state and always compare the results of GTP and GDP simulations. It is likely that, by considering relative changes, some of these artifacts cancel out and their effect on the conclusions is smaller than it would be for absolute values. Therefore, the periodic and finite size effects described above are unlikely to significantly affect the conclusions. Overall, we assume that our PF models sufficiently accurately describe the dynamics of straight MT lattice regions distant from the dynamic MT tip, with the important simplification that intra-lattice correlations are restricted to the lateral dimension.

## Methods

### Force-field parameters and protonation states

The CHARMM22* force field [56] and the CHARMM-modified TIP3P water model [57] were used in all simulations. GTP/GDP parameters were adapted from those for ATP/ADP implemented in the CHARMM22/CMAP force field [57, 58]. Titration curves of histidines were calculated using the GMCT package [59] and assigned as described previously [17].

### Simulation system preparation and cryo-EM refinement

Initial models for the tubulin dimers were obtained from PDB IDs 3JAT (GMPCPP) and 3JAS (GDP) [21] by extracting the central dimer from the $3 \times 2$ lattice patches (chains A and H in the original PDBs). GMPCPP was converted into GTP by replacing the carbon atom between $\alpha$- and $\beta$-phosphate with an oxygen atom. The missing loop in the $\alpha$-subunit (residues 38-46) was modelled in for structure consistency using MODELLER version 9.17 [60] but excluded from further refinement. Unlike in our previous study [17], we did not include the missing C-termini ($\alpha$:437–451 and $\beta$:426–445) in our simulations to reduce the system size and reach the best possible sampling. Unless differently specified, all structure and map manipulations were performed using UCSF Chimera [61] or VMD [62].

In all refinement simulations, the following data sets were used: EMD-6352 and EMD-6353 for symmetrized cryo-EM reconstructions of 14-PF GMPCPP- and GDP-MTs decorated with kinesin [21]. To create 'infinite' single-, double-, and three-PF systems, where the actual simulated part comprises exactly one layer of dimers and is coupled to copies of itself through axial periodic boundaries, we first constructed finite PF systems comprising two layers of dimers in the axial direction. To this end, subsections of the cryo-EM maps with the desired PF topology were extracted using an orthorhombic box, and the single dimer models were rigid-body fitted into the PF maps. The constructed PF systems were solvated in a triclinic water box of size $8.0 \times 8.0 \times 22.0$ nm$^3$ (single-PF), $12.7 \times 12.7 \times 22.0$ nm$^3$ (double-PF), or $19.0 \times 19.0 \times 22.0$ nm$^3$ (three-PF). The systems were then neutralized with 150mM KCl.

Refinement was done with correlation-driven molecular dynamics implemented as a custom module in the GROMACS 5.0.7 package [63], following our previously published protocols [31]. Briefly, we used the cold-fitting protocol with the longest refinement time (*i.e.* T = 100 K and total run time of 50 ns) followed by 15 ns of simulated annealing. The starting values for the biasing strength and the simulated map resolution were set to $1 \times 10^5$ kJ mol$^{-1}$ and 0.6 nm and linearly ramped to $5 \times 10^5$ kJ mol$^{-1}$ and 0.2 nm, respectively. The quality of the resulting models and the goodness of fit were ensured by calculating common stereochemical and correlation metrics (Table A in S1 Material).

## MD simulations

The finite PF models were converted into 'infinite' PF models by removing the extra tubulin monomers and nucleotides. Water and ion atoms were then trimmed to conform to the experimental value of the axial periodic dimension $L_z$, namely, 8.31 nm for GMPCPP-MTs and 8.15 nm for GDP-MTs [21]. The number of ions in the trimmed water shell was fixed such as to keep the systems neutral and to maintain the ionic strength of 150mM KCl. All subsequent MD simulations were carried out with GROMACS 5.0.7 [63]. Lennard-Jones and short-range electrostatic interactions were calculated with a 0.95-nm cutoff, while long-range electrostatic interactions were treated using particle-mesh Ewald summation [64] with a 0.12-nm grid spacing. The bond lengths were constrained using the LINCS algorithm [65] (hydrogen bonds during equilibration and all bonds in the production runs). Velocity rescaling [66] with a heat bath coupling constant of 0.5 ps was used to control the temperature for solute and solvent separately. Applying virtual site constraints [67] allowed us to increase the integration step size to 4 fs in the production runs. Center-of-mass correction was applied to solute and solvent separately every 100 steps.

With the above parameters fixed, the equilibration protocol consisted of the following steps: (i) energy minimization using steepest descent; (ii) short NVT equilibration for 1ns at T = 100 K with position restraints on heavy atoms and using a 1-fs integration time step; (iii) gradually heating up the system to 300 K within 10 ns in the NPT ensemble (Berendsen barostat [68] with a 5-ps coupling constant) using a 2-fs integration time step; (iv) equilibration in the NPT ensemble for 30 ns using isotropic Parrinello-Rahman barostat [69] with a 5-ps coupling constant and using a 2-fs integration time step; (v) equilibration in the NPT ensemble for 100 ns using semi-isotropic Parrinello-Rahman barostat with a 5-ps coupling constant and using a 2-fs time step. The last frame of step (v) was used to spawn stress-free production runs, stress-strain calculations, and umbrella sampling simulations.

## Derivation of the reaction coordinate

We carried out 20 independent, 1-$\mu$s long equilibrium simulations of the single-PF system in GTP-state, where the starting structure for each simulation was drawn every 150 ns from a

 

'seeding' simulation trajectory of 3 $\mu$s. For the single-PF system in GDP-state, we carried out 10 independent simulations (1 $\mu$s each) with the starting configurations drawn every 300 ns from a 3-$\mu$s 'seeding' trajectory. We then extracted backbone atoms ($N$, $C_\alpha$, $C$ and $O$) and excluded flexible protein regions ($\alpha$: 38-46, $\alpha$: 278-284 and $\beta$: 276-284) from further analysis.

Partial least-squares (PLS) functional mode analysis [33, 34] was then applied to the combined simulation set (both GTP- and GDP-state) to derive the collective mode of motion that correlated best with the fluctuations of the axial periodic dimension $L_z$ and had the largest variance in terms of molecular motion. The linear regression model was trained on the first half of the GTP data set ($\sim$ 13 $\mu$s) and the second half of the GDP data set ($\sim$ 7 $\mu$s), and the remaining halves were used for cross-validation. The cross-validation revealed that the ensemble-weighted collective mode (corresponds to the solution with one PLS component by construction) had correlation coefficients of 0.9 (training set) and 0.85 (validation set), hence yielding a robust representation of the conformational transition between the expanded GTP- and compacted GDP-state (Fig 2B). A visualization of this transition is shown in a supplementary movie (see S1 Material).

### Normalized mutual information and confinement entropy

In theory, the mutual information (MI) between two stochastic quantities $\chi_1$ and $\chi_2$ is

$$I(\chi_1, \chi_2) = H(\chi_1) + H(\chi_2) - H(\chi_1, \chi_2), \tag{1}$$

where $H(\chi_i) = - \int p_i(\chi_i) \log p_i d\chi_i$ is the entropy and $p_i(\chi_i)$ is the probability density of $\chi_i$ ($i = 1$, 2). The joint entropy $H(\chi_1, \chi_2)$ is defined similarly and requires knowledge of the joint probability density $p_{12}(\chi_1, \chi_2)$.

In practice, calculation of the MI is very sensitive to how the underlying probability densities are discretized. Too coarse-grained discretization leads to an underestimation and too detailed discretization leads to an overestimation of the MI. We therefore used the Jack Knifed estimate that is known to be a low bias estimate of the MI and robust to discretization bin size [70]. It is defined by substituting the entropy in Eq 1 with the following estimate $\hat{H}_{JK}(\chi_i) = N\hat{H}(\chi_i) - \frac{N-1}{N} \sum_{j=1}^{N} \hat{H}_{-j}(\chi_i)$, where $\hat{H}(\chi_i)$ is the entropy calculated by a straightforward discretization and $\hat{H}_{-j}(\chi_i)$ is the same as $\hat{H}(\chi_i)$ but when leaving out bin value $j$, and $N$ is the total number of bins. The confinement entropy used to estimate the conformational space 'volume' is then $H_{\text{conf}} \equiv \hat{H}_{JK}(\chi_1, \chi_2)$, whereas the normalized mutual information is defined as:

$$\text{NMI}_{12} = \frac{\hat{H}_{JK}(\chi_1) + \hat{H}_{JK}(\chi_2) - \hat{H}_{JK}(\chi_1, \chi_2)}{\sqrt{\hat{H}_{JK}(\chi_1)\hat{H}_{JK}(\chi_2)}}. \tag{2}$$

### Stress-strain simulations of isolated PFs

To measure the response of the single-PF systems to external axial strain, we let the prepared systems equilibrate under anisotropic pressure conditions $P_{xx} = P_{yy} \neq P_{zz}$ until convergence of $L_z$, where $P_{zz}$ ranged from $-65$ bar to $+200$ bar. All equilibration simulations were run for at least 1 $\mu$s, and the last 200 ns were used for further analysis. Due to the pressure difference maintained by the barostat, the simulated system (both solute and solvent) was subjected to an axial force $f_z$ such that the net stress on the PF along the $z$-axis, $\sigma_{zz}$, is:

$$\sigma_{zz} = -\frac{f_z}{A_z} = -\frac{(P_{zz} - P_\perp)L_x L_y}{A_z}, \tag{3}$$

 

where $P_\perp = (P_{xx} + P_{yy})/2$, $L_x$ and $L_y$ are the lateral dimensions of the simulation box, and $A_z$ is the PF cross-section area (see next section). The axial strain was computed as:

$$\varepsilon_{zz} = \frac{L_z - L_{z,eq}}{L_{z,eq}}, \tag{4}$$

where $L_{z,eq}$ is the mean axial periodic dimension. We also note that $P_{zz}$, $P_\perp$, and $L_{x,y,z}$ are, generally speaking, stochastic quantities. Therefore, block averaging with five blocks per trajectory and basic error propagation rules were used to estimate the mean and standard deviation of $\varepsilon_{zz}$ and $\sigma_{zz}$.

## Calculation of per-dimer elastic strain energy and flexural rigidity

According to linear elasticity theory, the per-dimer energy stored in a GTP-PF subjected to an axial elastic deformation by work required to compress it to the equilibrium state of a GDP-PF is $\Delta G_{el} = \Delta g_{el} V$, where $\Delta g_{el}$ is the elastic energy density and $V$ is the effective dimer volume. Using the generalized Hooke's law, we estimated the elastic energy density as $\Delta g_{el} = \frac{1}{2} E_{GTP} \varepsilon_{zz}^2$ in which $E_{GTP} \approx 0.89$ GPa (see Fig 2 in the main text) and $\varepsilon_{zz} = (L_{z,eq}^{GTP} - L_{z,eq}^{GDP})/L_{z,eq}^{GDP} \approx 0.03$ is the axial strain tensor component reflecting the difference in the equilibrium dimer lengths of GTP- and GDP-PFs (derived from stress-free simulations). We then calculated the effective dimer volume by requiring that the PF cross-section area $A_z$ matches the mass per PF unit length $m$, i.e. $m = \rho A_z L_{z,eq}$, where $m \approx 100$ kDa and $\rho \approx 1.41$ g/cm$^3$ is the mass density of globular proteins with molecular weights $M > 30$ kDa [71]. This yielded $A_z \approx 14.2$ nm$^2$, which allowed us to directly compute the sought elastic strain energy $\Delta G_{el} \approx 28.9$ kJ/mol $\approx 11.6$ $k_B$T at $T = 300$ K.

Following previous work [26], the flexural rigidity (or bending stiffness) of a long hollow cylindrical filament is a product of its axial elasticity modulus $E$ and the second moment of the cross-sectional area $I = \frac{\pi}{4}(R_{out}^4 - R_{in}^4)$, where $R_{in}$ and $R_{out}$ are the inner and outer radii of the cylinder, respectively. Using the estimate $R_{in} \approx 10.19$ nm from [72] (for 14_3 type MTs) and requiring that the MT cross-section conforms with the mass per MT unit length, i.e. $14 \times m = \rho \pi (R_{out}^2 - R_{in}^2) L_{z,eq}$, we obtained an estimate for the outer radius $R_{out} \approx 12.95$ nm and, hence, for the second moment $I \approx 1.31 \times 10^{-32}$ m$^4$. This value allowed us to directly compute the flexural rigidities of GTP- and GDP-MTs (see Fig 2 in the main text).

## Estimating MT bending stiffness from previous experimental data

The experimental values for MT bending stiffnesses and the respective uncertainties shown in Fig 2D were calculated using inverse-variance weighting [73]. Given a set of independent measurements $y_i$ with variances $\sigma_i^2$, the consensus inverse-variance mean and standard deviation are given by $\hat{y} = \sum_i w_i y_i / \sum_i w_i$ and $\hat{\sigma} = \sqrt{1/\sum_i w_i}$, where the weights $w_i = 1/\sigma_i^2$. For GDP-MTs and GDP-MTs stabilized with Taxol, we used the $E \times I$ values estimated by quantifying thermal fluctuations of MTs, as summarized in [29] and [30]. As there were only few measurements of GMPCPP-MTs in the cited publications, we extended the set by considering further thermal fluctuation studies [46, 47, 74].

## Code and supplementary data

All refined starting structures are provided as Supplementary Data Sets. Unless explicitly specified, all numerical calculations were carried out using Python 2.7 [75] and Cython [76].

## Supporting information

**S1 Material. Supplementary text that includes all supplementary figures and tables as well as detailed information on stress-strain calculations, estimation of the per-dimer elastic strain energy, umbrella sampling simulations, estimation of the relative lateral bond stability, and Bayesian inference of the joint free energy distribution for the three-PF system.**
(PDF)

**S1 Movie. Animation showing the compaction transition derived from equilibrium simulations of the single-PF system in both GTP- and GDP-state (see Fig 2).**
(MOV)

**S1 Data Set. Archive containing refined single-, double- and three-PF structures in both GTP- and GDP-state.** S1: Refined single-PF structure in the GTP state. S2: Refined single-PF structure in the GDP state. S3: Refined homotypic double-PF structure in the GTP state. S4: Refined homotypic double-PF structure in the GDP state. S5: Refined three-PF structure in the GTP state. S6: Refined three-PF structure in the GDP state.
(ZIP)

## Acknowledgments

We thank Rui Zhang (Washington University in St. Louis, USA) and Eva Nogales (UC Berkeley, USA) for insightful discussions and for kindly providing the microtubule cryo-EM reconstructions; Gregory Bubnis (UC San Francisco, USA) and Thomas Ullmann (MPI-BPC, Göttingen, Germany) for suggestions about free energy calculations and error estimation. Computational resources were provided by the North-German Supercomputing Alliance (Berlin/Göttingen, Germany) as well as by the Max Planck Computing and Data Facility and the Leibniz Supercomputing Centre (Garching, Germany).

## Author Contributions

**Conceptualization:** Maxim Igaev, Helmut Grubmüller.

**Data curation:** Maxim Igaev.

**Formal analysis:** Maxim Igaev.

**Funding acquisition:** Maxim Igaev, Helmut Grubmüller.

**Investigation:** Maxim Igaev.

**Methodology:** Maxim Igaev, Helmut Grubmüller.

**Project administration:** Maxim Igaev.

**Resources:** Helmut Grubmüller.

**Software:** Maxim Igaev.

**Supervision:** Helmut Grubmüller.

**Validation:** Maxim Igaev.

**Visualization:** Maxim Igaev.

**Writing – original draft:** Maxim Igaev.

**Writing – review & editing:** Maxim Igaev, Helmut Grubmüller.

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
