## [Decision Letter · Decision Letter 0]

30 Apr 2020

Dear Dr. Igaev,

Thank you very much for submitting your manuscript "Microtubule instability driven by longitudinal and lateral strain propagation" for consideration at PLOS Computational Biology.

As with all papers reviewed by the journal, your manuscript was reviewed by members of the editorial board and by several independent reviewers. In light of the reviews (below this email), we would like to invite the resubmission of a significantly-revised version that takes into account the reviewers' comments.

We cannot make any decision about publication until we have seen the revised manuscript and your response to the reviewers' comments. Your revised manuscript is also likely to be sent to reviewers for further evaluation.

Sincerely,

Turkan Haliloglu

Associate Editor

PLOS Computational Biology

Nir Ben-Tal

Deputy Editor

PLOS Computational Biology

Reviewer's Responses to Questions

**Comments to the Authors:**

Reviewer #1: Please see attached file

Reviewer #2: This is an MD-based study on the conformational and energetic consequences of GTP hydrolysis within and across protofilaments in microtubuli, starting from available cryo EM data. The data is very solid, based on long sampling, free energy and entropy calculations, and supports clearly the conclusions, with only one concern with regard to entropy vs free energy considerations, see below. The simulations definitely advance the field and support, by solid numbers, the mechanisms relevant for destabiliziation of MTs.

I fully recommend the study for publication in Plos Comp Biol.

My points:

Entropy calculations and free energy argument Fig. 3: The entropy and free energy calculations both are based on the conformational freedom with regard to dimer spacing (by definition the reaction coordinates in the umbrella sampling), if I am not mistaken. Why is this a good measure, or could there be other changes in free energy involved, such as PF spacing? Also, the NMI and free energies both point into the same direction, but the two fold higher NMI for GDP is used as a stability argument, without referring to the respective dG (which is the actual thermodynamic measure). Couldn’t one read off the free energy ‘penalty’ by confinement (beyond just the entropy) from the free energy maps? Is a mismatch of x causing an increase in free energy of y which is higher (two-fold? or lower?) for GDP compare to GTP? In other words, is the NMI measuring the important thing here?

This question also relates to a major conclusion of the paper: “While not significantly affecting lateral bond stability, the stored elastic energy results in more strongly confined and correlated dynamics of GDP-tubulins, thereby entropically destabilizing the MT lattice.”

It is only entropy? Or could lateral interactions also play a role? This does not seem to be sufficiently quantified by showing that the umbrella sampling and the NMI values point into similar directions.

Given that the difference between GTP and GDP are roughly 0.2nm, it is a bit concerning that the simulation deviate on average from the cryo structures by 0.1nm (Figs 3b,c), luckily both to a similar extent, so that the mismatch remains the same. Still, what could be a reason? This should be discussed.

Minor remarks:

“We hence compared our calculations only with thermal fluctuation experiments because non-elastic deformations as well as induced contact breaking are less likely to occur under such conditions, consistent with our small-strain simulations”

This sentence is misleading. Even if non-elastic deformation are likely in the thermal fluctuation experiments (can not be excluded?), they are essentially at zero strain (or zero-to-small strain) and thus close to your small-strain simulations, that is what you want to say?

Fig 2d: why using the negative stress values and not the positive ones for calculating bending stiffness? Is that common sense? Bending includes both, pulling on one side and pushing on the other, so both elastic moduli (a mean?) could contribute?

“Pure GMPCPP-MTs and GDP-MTs differ in dimer spacing but are homogeneous in their structure and dynamics, because they consist of mechanochemically equal dimers that explore roughly the same conformational space.”

Again misleading: how can the conformational space be the same (of the MTs not only the dimers), but the dimer spacing is different? Would the latte rnot imply different conformational space?

Fig 3b: Please add the meaning of the 0.19nm to caption, just looking at the plots (and searching in the text) eventually resolves it, but it would still help the reader.

Very minor points:

“in a MT” -> in an MT rather?

“father away”

Reviewer #3: This well-written manuscript describes a molecular dynamics study of microtubule energetics. Using recent high-resolution cryo-EM structures, the authors construct effectively infinite models of microtubule filaments in two chemical states corresponding to the initial and the final points of a GTP hydrolysis reaction. Extensive simulations of two protofilament systems revealed a considerable difference in the amplitude of equilibrium fluctuations, indicating a difference in the elasticity of the filament in the two chemical states. The mechanical properties were directly determined by simulating the two systems at various values of the axial pressure tensor. Further analysis of the simulation results produced estimates of the free-energy stored in the filament structure upon GTP hydrolysis, which was found to be in good agreement with the corresponding experimental estimates. The authors next investigate the energetics of filament dimers and trimers through a combination of umbrella sampling simulations and theoretical considerations. The key findings include the free energy cost of lateral incorporation a mismatched tubulin dimer and the degree of correlation in a microtubule lattice.

Overall, this is an exemplary study that demonstrates the power of high-end molecular dynamics simulations in not only examining a qualitative behavior of a biomolecular system but also in providing quantitative estimates of the system’s energetics, including the uncertainty of such determination. While the study is not free from possible artifacts, such as the use of an effectively infinite system for the study of local protein compaction in a dimer filament system, the expected effect of such artifacts is adequately described in the Discussion section of the manuscript. Specific to the microtubule field, the results of the study provide much needed microscopic perspective on the energetics of microtubule growth, setting the stage to future exascale simulations.

**Have all data underlying the figures and results presented in the manuscript been provided?**

Reviewer #1: Yes

Reviewer #2: Yes

Reviewer #3: Yes

PLOS authors have the option to publish the peer review history of their article (what does this mean?). If published, this will include your full peer review and any attached files.

Reviewer #1: No

Reviewer #2: No

Reviewer #3: Yes: Aleksei Aksimentiev
---

## [Decision Letter · Decision Letter 1]

29 Jun 2020

Dear Dr. Igaev,

Thank you very much for submitting your manuscript "Microtubule instability driven by longitudinal and lateral strain propagation" for consideration at PLOS Computational Biology. As with all papers reviewed by the journal, your manuscript was reviewed by members of the editorial board and by several independent reviewers. The reviewers appreciated the attention to an important topic. Based on the reviews, we are likely to accept this manuscript for publication, providing that you modify the manuscript according to the review recommendations.

Sincerely,

Turkan Haliloglu

Associate Editor

PLOS Computational Biology

Nir Ben-Tal

Deputy Editor

PLOS Computational Biology

[LINK]

Reviewer's Responses to Questions

**Comments to the Authors:**

Reviewer #1: Please see attached.

Reviewer #2: My points have been addressed with care. I could not spot the differently colored text in the revised version though

**Have all data underlying the figures and results presented in the manuscript been provided?**

Reviewer #1: Yes

Reviewer #2: Yes

PLOS authors have the option to publish the peer review history of their article (what does this mean?). If published, this will include your full peer review and any attached files.

Reviewer #1: No

Reviewer #2: No
---

## [Editor Report · Decision Letter 2]

9 Jul 2020

Dear Dr. Igaev,

We are pleased to inform you that your manuscript 'Microtubule instability driven by longitudinal and lateral strain propagation' has been provisionally accepted for publication in PLOS Computational Biology.

Best regards,

Turkan Haliloglu

Associate Editor

PLOS Computational Biology

Nir Ben-Tal

Deputy Editor

PLOS Computational Biology

---

## [Editor Report · Acceptance letter]

7 Aug 2020

PCOMPBIOL-D-20-00379R2 

Microtubule instability driven by longitudinal and lateral strain propagation

Dear Dr Igaev,

I am pleased to inform you that your manuscript has been formally accepted for publication in PLOS Computational Biology. Your manuscript is now with our production department and you will be notified of the publication date in due course.

With kind regards,

Laura Mallard
